# HiResNets: Native Full-HD Video Recognition with Foveal Residual Streams

## 1 Abstract

Much of the recent progress in image and video recognition has come at the cost of memory: larger models, increased resolution, and longer temporal contexts. An inevitable component is the quadratic (or larger) growth of memory and compute based on image resolution, which is a property of the grid sampling used in convolutional networks and vision transformers. In this work we study residual networks whose convolutional blocks have logarithmic-square growth instead, enabling them to process very high-resolution video quickly and with low memory. The key insight is to use a residual architectures' residual stream as a high-resolution buffer, to which convolutional blocks only read and write via log-polar image warp operations. Layers adaptively focus on different parts of each frame, with very high resolution only near the focus point. A complete high-resolution representation is built up in the residual stream, which is analogous to eye saccades creating a complete picture in biological vision. Experiments demonstrate that our proposed HiResNets learn to foveate around scenes similarly to human vision, and have superior performance in difficult egocentric video recognition tasks, especially egocentric video with small objects and fine-grained recognition.

## 2 Introduction

Progress in image and video recognition has been driven by ever-larger models, higher input resolutions, and longer temporal contexts. This trend has clear costs: memory and compute grow at least quadratically with spatial resolution in convolutional networks and vision transformers (Dosovitskiy et al., 2021; He et al., 2016), creating a hard bottleneck for tasks that require fine detail. Applications such as egocentric video recognition or small-object analysis are especially constrained, not because of a lack of model capacity, but because of the inefficiency of uniform grid-based sampling.

In contrast, the human eye allocates resolution unevenly, capturing detail only at the fovea while encoding the periphery coarsely, and relies on saccades to integrate a full high-resolution scene (Yamada et al., 2018). This principle has motivated several works: from earlier glimpse networks (Mnih et al., 2014), zoom-in detectors (Cao et al., 2018), and hierarchical multi-scale processing (Larochelle & Hinton, 2010), to more recent approaches which incorporated saccade-like glimpses into modern architectures, for example through recurrent hard-attention models (Elsayed et al., 2019; Li et al., 2025) or differentiable foveated sampling schemes (Deza & Konkle, 2023). Despite these advances, the common limitation is that foveation is applied outside the backbone itself (typically as a control or pre-processing module wrapped around an otherwise standard network), so the backbone continues to scale quadratically with resolution.

We address this limitation by embedding foveation directly into the backbone. Our key idea is to use the residual pathway of a deep network as a persistent high-resolution buffer, while convolutional blocks interact only with a warped, adaptive-resolution view. This view is produced by a log-polar image warp, which preserves fine detail near a chosen focus point and compresses the periphery (Schwartz, 1980). As a result, the cost of residual blocks grows only logarithmically with resolution, rather than quadratically, while the residual stream maintains full fidelity. Layers can shift their focus adaptively across frames, gradually building up a complete high-resolution representation in a manner analogous to biological saccades.

While this means that the residual stream still scales quadratically with resolution, it is only involved in inexpensive copy and addition operations, and its back-propagation memory can be reduced (for example with gradient checkpointing (Chen et al., 2016)). We thus have greater performance gains

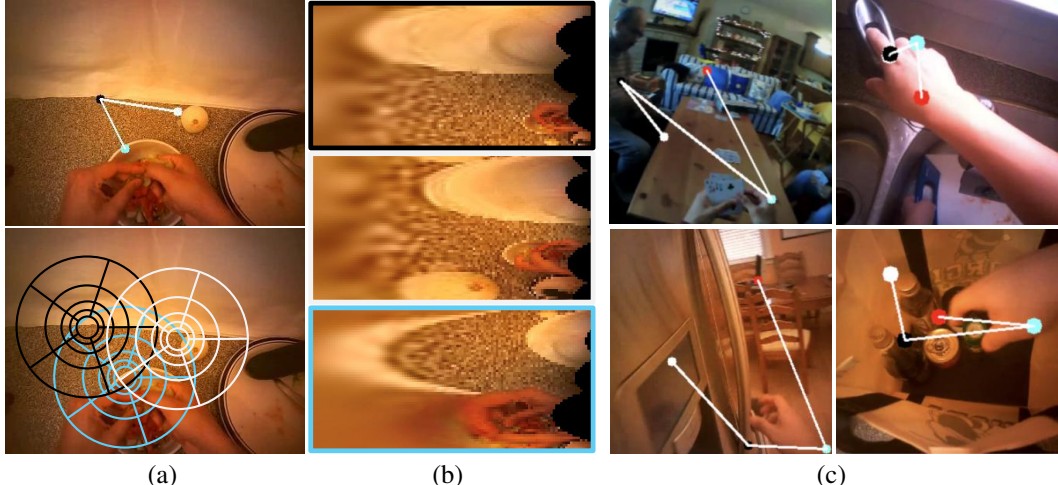

(a)                  (b)                  (c)

Figure 1: Illustration of the foveal warps and qualitative results of gaze estimation. (a) Three gazes estimated by different stages of the network (starting with the white dot, top), and the corresponding log-polar warp grids (bottom) (sec. 4.2). (b) Illustration of log-polar warped images (in practice the residual blocks operate on residual features, not raw RGB images). (c) Examples of saccade trajectories that appear in egocentric video. The network tends to focus on hands in close-up scenes (important for action prediction and locating manipulated objects), and alternating near-far regions in panoramic scenes.

by improving the residual blocks, which are often the bottleneck due to their expensive convolutions and expanded numbers of channels (Liu et al., 2022).

The resulting architecture, **HiResNets**, integrates foveation directly into residual networks. Unlike prior approaches that wrap a standard backbone with glimpse or zoom modules, HiResNets modify the internal computation of each residual block, yielding fundamentally different scaling behaviour with respect to input resolution. Our contributions are threefold:

1. A residual design in which block cost grows only logarithmically with resolution while the residual stream maintains full fidelity.

2. A differentiable log-polar warp mechanism enabling adaptive foveated processing inside the backbone itself.

3. Extensive experiments showing that HiResNets not only reduce memory and computation but also learn interpretable foveation strategies.

On egocentric video benchmarks, HiResNets offers consistent gains, particularly for fine-grained activities and small-object recognition. These results demonstrate that foveated architectures can overcome resolution bottlenecks in video understanding.

## 3 RELATED WORK

There is an ample body of literature that is related to our work. In this section we provide only a small summary of more recent and classical papers that are relevant.

**Biological Vision and Foveation.** It is well known that human vision is sharply non-uniform: the retina contains a densely packed fovea surrounded by coarse peripheral sampling, with retinotopic organization that approximates a log-polar map. Classic studies of visual attention emphasize that this arrangement is computationally efficient, as attention enhances acuity only where it is needed (Carrasco, 2011). Anatomical evidence confirms dramatic variation in photoreceptor density across the retina (Curcio et al., 1990), and early computational accounts framed retinotopy itself as a log-polar coordinate transform (Schwartz, 1977). Perceptual research has shown how peripheral vision

is limited by crowding and coarse feature integration (Strasburger et al., 2011), while summary-statistic representations explain how such limits still support robust search and scene perception (Rosenholtz et al., 2012). Yamada et al. (2018) demonstrated engineering applications of foveated vision systems, showing how uneven resolution sampling can reduce bandwidth and computation. These works show that foveation and saccades evolved in biology as an efficient strategy for building complete high-resolution representations from partial glimpses, as opposed to having a uniform-resolution sensor, which is commonly the case in artificial vision.

**Log-Polar Compression and Differentiable Warping.** This biological perspective has inspired computer vision models that explicitly encode non-uniform sampling. Focusing on the deep learning era, early work in geometric warping introduced learnable modules such as spatial transformer networks (Jaderberg et al., 2015), while polar transformer networks showed how polar coordinates yield built-in rotation and scale equivariance (Esteves et al., 2018b). More recently, log-polar convolution layers were proposed to natively operate in a retinotopic coordinate system, yielding both efficiency and robustness to geometric variation (Su & Wen, 2022).

**Computational Models of Foveation.** Beyond static reparameterizations, many learning-based systems attempt to mimic saccades by dynamically selecting where to process at high resolution. The Recurrent Model of Visual Attention (RAM) introduced sequential glimpses trained via reinforcement learning (Mnih et al., 2014), while later models such as Saccader stabilized accuracy by supervising fixation selection (Elsayed et al., 2019). Other approaches replaced reinforcement learning with differentiable mechanisms: the Dynamic Zoom-In network, for example, predicted where to zoom within large images to save computation (Gao et al., 2018). Cao et al. (2018) introduced zoom-in detection pipelines where a coarse network proposes candidate regions that are re-examined at higher resolution, reducing cost for large images. More recent methods have implemented continuous foveated sensors that sample the input with log-polar density and learn how to shift fixations end-to-end (Killick et al., 2023), or incorporated foveation directly into transformers, as in FoveaTer (Jonnalagadda et al., 2021). Monte Carlo convolutions generalize filtering to non-uniform foveated inputs (Killick et al., 2022), and multi-resolution strategies such as Dragonfly achieve similar goals by aggregating many zoomed sub-crops into a unified representation (Thapa et al., 2024). Recently, Li et al. (2025) proposed MRAM, a multi-level recurrent attention model that mimics fixations and saccades to improve stability and accuracy in glimpse-based architectures.

**Egocentric Vision and Gaze Estimation Benchmarks.** Egocentric video is a natural application domain for foveated models, since hand-object interactions, rapid egomotion, and small tools make uniform downsampling especially lossy. Benchmarks such as EPIC-KITCHENS (Damen et al., 2018) and Ego4D (Grauman et al., 2022) established large-scale testbeds for activity recognition and object understanding, while HD-EPIC (Perrett et al., 2025) recently added highly detailed annotations and gaze data. In parallel, gaze-estimation datasets such as MPIIGaze (Zhang et al., 2019), ETH-XGaze (Zhang et al., 2020), and Gaze360 (Kellnhofer et al., 2019), along with VR-focused corpora like OpenEDS (Garbin et al., 2019), provide evidence of where humans naturally focus in first-person settings.

**Small-Object Detection and Fine-Grained Targets.** Standard detection backbones lose fine detail under pooling and stride, which led to several multi-scale architectures. Larochelle & Hinton (2010) presented one of the earliest hierarchical multi-scale models, showing that learning across resolutions improves recognition efficiency. Feature Pyramid Networks (Lin et al., 2017a) explicitly added top-down pathways to preserve detail across scales, while RetinaNet (Lin et al., 2017b) introduced focal loss to mitigate the imbalance between small and large objects. Transformer-based detection, exemplified by DETR (Carion et al., 2020), simplified pipelines but still struggled on small objects without scale-specific augmentation.

**Large-Scale High-Resolution Vision.** Finally, a broad literature addresses how to scale vision architectures to high-resolution images and video. HRNet demonstrated the benefits of maintaining parallel high-resolution streams throughout a network (Sun et al., 2019; Wang et al., 2020), while Multiscale Vision Transformers (MViT) and its successor MViTv2 used hierarchical pooling to manage compute (Fan et al., 2021; Li et al., 2022). Efficiency-focused transformers prune or merge tokens dynamically, as in DynamicViT (Rao et al., 2021), EViT (Liang et al., 2022), and ToMe (Bolya

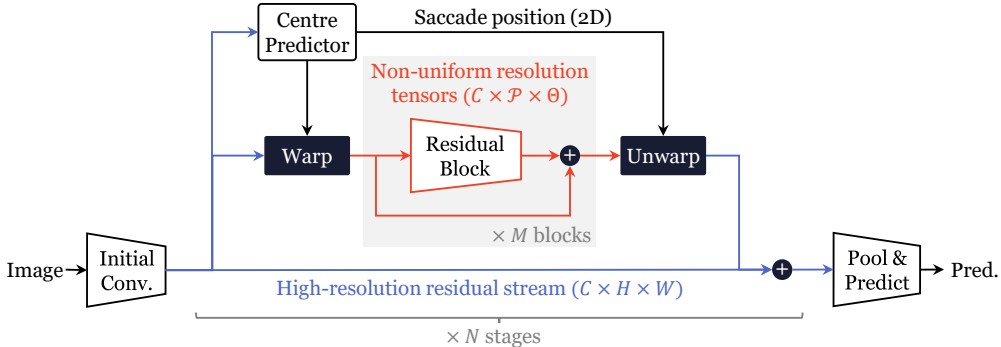

Figure 2: Overview of the proposed architecture. We integrate log-polar warp and unwarp operators around a series of residual (convolutional) blocks of a given backbone (e.g. ResNet). In this warped space, the network can process a very high resolution only around a saccade position (predicted by a separate branch), with much reduced computation. This allows the residual stream to carry information at a much higher resolution, processing full-HD video natively and recognizing very fine-grained detail.

et al., 2022). At the extreme end, gigapixel pathology has motivated hierarchical pretraining (HIPT) (Chen et al., 2022) and MIL-based slide classification (CLAM) (Lu et al., 2021). These proposals attempt to balance resolution with efficiency, but most operate at globally fixed scales or discard fine detail. In contrast, the proposed HiResNets achieve sub-quadratic (log-squared) scaling by treating the residual stream itself as a high-resolution buffer accessed via log-polar warps, allowing full-HD video to be processed natively while preserving the biological analogy of foveation and saccades.

## 4 METHOD

### 4.1 RESIDUAL NETWORKS AND RESOLUTION SCALING

We begin by recalling the standard residual block notation of He et al. (2016). Let $x^{(l)} \in \mathbb{R}^{C \times H \times W}$ denote the feature map (the residual stream) after block $l$. A residual block updates it via

$$x^{(l+1)} = x^{(l)} + f\left(x^{(l)}\right), \tag{1}$$

where $f(\cdot)$ is a sequence of convolutions, nonlinearities, and normalizations.

While this formulation underlies most modern vision models, its cost grows quickly with resolution. The computational complexity of a convolution with kernel size $K$ is

$$\mathcal{O}(C^2 H W K^2), \tag{2}$$

so both compute and memory scale quadratically in the spatial dimensions. This quadratic law is the main bottleneck preventing standard residual networks from handling very high-resolution video.

### 4.2 LOG-POLAR WARPS

To alleviate this bottleneck, we require a representation that emphasizes local detail while compressing distant regions. One candidate is the log-polar reparameterization of the image plane (Esteves et al., 2018a). Let $c \in \mathbb{R}^2$ be a centre of focus. The log-polar mapping $\psi_c : \mathbb{R}^{\mathcal{P} \times \Theta} \to \mathbb{R}^{H \times W}$ is defined by

$$(i, j) = \psi_c(\rho, \theta) = c + \big(\exp(\rho)\cos\theta, \ \exp(\rho)\sin\theta\big), \tag{3}$$

where $(\rho, \theta)$ are the log-polar coordinates of a point in the warped view. This mapping allocates exponentially more resolution near $c$, while progressively downsampling farther away.

The warped feature at $(\rho, \theta)$ is obtained by bilinear resampling:

$$v(\rho, \theta) = \sum_m \sum_n x_{m,n}\, \varphi(i - m)\, \varphi(j - n), \qquad \varphi(t) = \max(0, 1 - |t|). \tag{4}$$

Since $\varphi$ has compact support in $[-1, 1]$, this double sum reduces to exactly four terms: the 4 neighbours obtained by integer floor and ceiling of the coordinates.

**Limitations of direct warping.** Although the log-polar warp provides a more efficient parameterization, applying it directly to the input image or to the activations of a standard network has a major limitation: the centre $c$ is fixed for the entire forward pass. As a result, only a single part of the scene benefits from high-resolution processing, and the network cannot reallocate its resolution budget across layers or time. This motivates applying the warp at a more fine-grained level.

### 4.3 HiResNet architecture

We therefore restructure the residual block so that convolutional processing takes place only in log-polar space, while the global high-resolution representation is preserved in the residual stream. Concretely, a block is defined as

$$u = \psi_c(x), \qquad y = f(u), \qquad x \leftarrow x + \psi_c^{-1}(y), \tag{5}$$

where $\psi_c^{-1}$ is the inverse log-polar warp, mapping features back to Cartesian coordinates by bilinear resampling.

This means that each convolutional block operates on a compact warped view $u \in \mathbb{R}^{C \times \mathcal{P} \times \Theta}$, while $x$ maintains a full-resolution buffer of the scene. Over the course of $N$ blocks, the network integrates multiple warped updates into $x$, analogous to how the visual system fuses multiple saccades into a coherent high-resolution percept. In practice, we let $f$ be a sequence of $M$ residual blocks instead of just one, in order to avoid recomputing the centre $c$ too often over the depth of the network.

### 4.4 Saccade predictor

To make this mechanism adaptive, the centre $c$ must be predicted dynamically at each block. We introduce a lightweight *saccade predictor*, consisting of two $1 \times 1$ convolutions with a ReLU nonlinearity, followed by a differentiable *softargmax* operator (Henriques & Vedaldi, 2017). Given an attention map $a \in \mathbb{R}^{H \times W}$, the softargmax produces

$$c = \sum_{i,j} (i, j) \frac{\exp(a_{i,j})}{\sum_{m,n} \exp(a_{m,n})}. \tag{6}$$

This allows each block to reposition its high-resolution focus based on the current residual state $x$.

### 4.5 Inverse warp

In order to relate high-resolution information from different focus positions, we must be able to write information back from a log-polar warped space into a common space (typically cartesian). We thus use also the following inverse warp to transform features to the high-resolution residual stream:

$$(\rho, \theta) = \psi_c^{-1}(i, j) = \left( \log \|(i', j')\|^2, \ \mathrm{atan2}(j', i') \right), \quad (i', j') = (i, j) - c, \tag{7}$$

again using bilinear interpolation (eq. 4).

### 4.6 Complexity analysis

A standard residual block with kernel size $k$ applied to a feature map of size $(W, H)$ has cost $\mathcal{O}(k^2 W H)$. In our proposal, convolutions act only on the warped view $u \in \mathbb{R}^{C \times \Theta \times \mathcal{P}}$, where the log-polar warp yields $\Theta = \mathcal{O}(\log W)$ and $\mathcal{P} = \mathcal{O}(\log H)$. The warp itself can be implemented in $\mathcal{O}(\Theta \mathcal{P})$, so the dominant cost is the convolution, scaling as

$$\mathcal{O}(k^2 \Theta \mathcal{P}) = \mathcal{O}(k^2 \log W \log H).$$

This logarithmic-square scaling replaces the quadratic growth of conventional blocks, and becomes the bottleneck whenever convolution dominates warp overhead and channel dimensions remain moderate.

Meanwhile, the residual stream ensures that no fine detail is lost: every update is reintegrated into a global high-resolution buffer. Thus HiResNets achieve logarithmic-square scaling in the computationally dominant convolutional bottlenecks, while preserving complete spatial information across depth and time.

### 4.7 Implementation details

In each experiment, we take an existing residual network as a baseline (ResNet (He et al., 2016) or SqueezeTime (Zhai et al., 2024)), and obtain a HiResNet by adding the warp operations as described in sec. 4.3. We implement the log-polar warp (sec. 4.2) and its inverse transform modules using efficient bilinear sampling (`grid_sample` in PyTorch) in order to produce the warped and unwarped tensors.

A new localization network is instantiated in every stage (group of residual blocks) – for example, the ResNet always has 4 stages (He et al., 2016). Therefore there is one focus point per stage. We also experiment with the frequency of the polar prediction, for block groups with large number of residual blocks.

As for spatial resolution of the tensors, the residual blocks' spatial sizes increase by 1/2 every stage with increasing depth, while the residual stream has a constant size 4 times smaller than the input resolution.

## 5 Experiments

In these experiments, we want to assess several capabilities of HiResNets: 1) their ability to mimic gaze estimation, analogously to biological vision; 2) the ability to process higher-resolution images and video than their corresponding baselines; 3) their performance scaling w.r.t. image resolution; 4) the ability to detect fine-grained object details that would be difficult in lower resolutions.

**Baselines.** We use YOLOv5 (Jocher et al., 2020) as a strong one-stage object detection baseline. It uses a CSPDarknet backbone with PANet feature aggregation and anchor-based detection heads, to trade off between accuracy and speed on high-resolution inputs. For gaze estimation, Global-Local Correlation (GLC) (Lai et al., 2023) is a transformer-based egocentric gaze estimation model that explicitly models the interaction between global scene context and local visual features. It injects a "global token" into the transformer embedding and uses a Global-Local Correlation (GLC) module to compute attention weights between that global token and every local token.

### 5.1 High-Resolution Egocentric Vision Experiments

We focus on egocentric video, which presents unique challenges that are relevant for our approach. Unlike third-person data, egocentric data is dominated by rapid head motion, cluttered environments, and frequent occlusions. Objects of interest are often small, hand-held, and viewed at unusual angles. These conditions make accurate recognition heavily dependent on preserving fine spatial detail, but requiring high resolution quickly becomes prohibitive. We therefore focus our experiments on egocentric object detection, where the ability of HiResNets to foveate adaptively and process high-resolution video efficiently is most critical.

In all these experiments, we evaluate the method at different resolutions, including relatively high ones (reaching Full HD resolution), in order to show the scaling behaviour of the different methods.

### 5.1.1 Gaze Estimation

Our first task is gaze estimation, which we speculate should have a highly correlated output with the foveation mechanism that we introduce. In egocentric gaze estimation, a model receives head-mounted first-person video and predicts the corresponding 2D gaze locations on each frame, indicating where the wearer is looking.

For datasets, we use Ego4D (Grauman et al., 2022) and EGTEA (Li et al., 2018). Ego4D contains 15K video clips for training and 5.2K video clips for testing, and 20K gaze trajectories. EGTEA contains 8.2K clips for training and 2K clips for testing, and 10K gaze trajectories. For all our gaze

| Method | Max Res | EGTEA | | | Ego4D | | |
|---|---|---|---|---|---|---|---|
| | | AP. | AR. | F1. | AP. | AR. | F1. |
| | 640 | 0.29 | 0.59 | 0.39 | 0.32 | 0.57 | 0.41 |
| SqueezeTime (Zhai et al., 2024) | 1000 | 0.33 | 0.60 | 0.42 | 0.34 | **0.58** | 0.42 |
| | 640 | 0.35 | 0.61 | 0.44 | 0.34 | 0.57 | 0.43 |
| MViT+GLC (Lai et al., 2023) | 1000 (OOM) | - | - | - | - | - | - |
| | 640 | 0.36 | 0.62 | 0.44 | 0.35 | 0.56 | 0.43 |
| Ours | 1000 | **0.37** | **0.63** | **0.46** | **0.36** | 0.57 | **0.44** |

Table 1: Gaze Estimation on EGTEA and Ego4D.

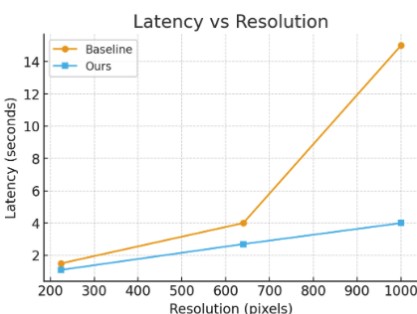

Figure 3: Computational cost plot (latency vs. resolution) on Ego4D (Grauman et al., 2022) gaze estimation task for the baseline (SqueezeTime) and our method.

estimation tasks, we integrate our method into the GLC (Lai et al., 2023) pipeline, including the native configuration of optimizer, batch sizes, and evaluation metrics (average precision, F1 and recall). We swap the video backbone architecture in GLC with SqueezeTime (Zhai et al., 2024) for both the baseline and the proposed method, since we found it to be more performant.

**Results.** We evaluate the performance when scaling the resolution from 640 to 1000. We observe that our model outperforms other baselines both at lower and higher resolutions (table 1). However, the task is largely solvable at intermediate resolutions, attributed to the distribution of the object sizes which contain both small and large objects.

### 5.1.2 OBJECT DETECTION – EGOOBJECTS AND EGO4D

Object detection is a classic task in computer vision and is one where focusing on different regions of an image seems like a natural strategy, since objects are bounded and some can be relatively small, especially in egocentric video.

For datasets, we use EgoObjects (Zhu et al., 2023) and Ego4D datasets (Grauman et al., 2022). EgoObjects is the largest egocentric object detection dataset, with 78K frames in training and 6K frames in validation, with 640K bounding boxes. We curate the Ego4D dataset with 53K frames in train and 35K frames in test and a total of 100K bounding boxes. We show the distribution of object sizes in both datasets. For all our object detection tasks, we use the state-of-the-art object detection model, YOLO (Jocher et al., 2020) with its native configuration of optimizer, batch size, and evaluation metrics (average precision, recall and evaluation accuracy).

**Results.** For Ego4D dataset, when scaling the resolutions, accuracy does not significantly improve as seen in table 2. This can be attributed to the distribution of the object sizes in the dataset which cover large pixel areas, hence do not need higher resolution training. In contrast, for EgoObjects, when scaling the resolutions, accuracy improves by about 10%, attributing to varied object size distribution from small to large pixel areas (table 2). However, the dataset is still largely solvable at intermediate resolutions as seen in the table. Varying improvements at increasing resolutions across

| Method | Max Res | Ego4D | | | Ego-Objects | | |
|---|---|---|---|---|---|---|---|
| | | AP. | AR. | Acc. | AP. | AR. | Acc. |
| YOLOv5 (Jocher et al., 2020) | 640 | 0.36 | 0.16 | 0.21 | 0.28 | 0.17 | 0.27 |
| | 900 | 0.37 | 0.15 | 0.23 | 0.26 | 0.18 | 0.29 |
| | 1200 | 0.35 | 0.18 | 0.25 | 0.27 | 0.16 | 0.28 |
| Ours | 640 | 0.41 | **0.24** | 0.27 | 0.35 | 0.20 | 0.37 |
| | 900 | **0.44** | 0.23 | **0.30** | **0.37** | 0.23 | **0.39** |
| | 1200 | 0.42 | 0.20 | 0.25 | 0.36 | **0.26** | 0.35 |

Table 2: Object Detection on Ego4D and Ego-Objects.

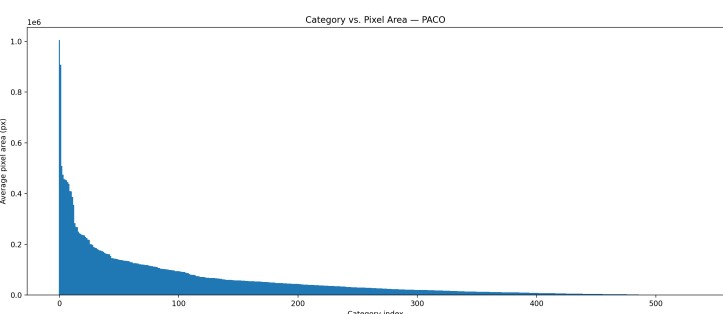

Figure 4: Average pixel area distribution for object and part categories in PACO (Ramanathan et al., 2023) dataset in its full resolution image.

the two datasets motivates us to investigate performance on a dataset primarily covering smaller objects sizes.

### 5.2 FINE-GRAINED VISUAL UNDERSTANDING

Finally, we turn to fine-grained visual understanding, for which we posit that high-resolution video understanding may be especially well-suited. We thus evaluate on the PACO (Ramanathan et al., 2023) detection dataset for detailed visual understanding. Derived from the Ego4D dataset (Grauman et al., 2022), PACO comprises of rich annotations 531 object categories, of which 456 are object-part categories. There are a total of 15667 frames in train and 550 frames in validation, and 197K bounding boxes. We plot the pixel area size distribution for the categories in the dataset in Fig. 4. Here, we also study the effectiveness of foveation by evaluating on dataset splits with varying object sizes (spatial area 0-5% and 5-10% of the image). While our method delivers stable evaluation performance for these splits, the baseline's accuracy degrades by 1-2% than that observed in table 3.

**Results.** We evaluate the performance on PACO when scaling resolution from 640 to 1400. We see that our method outperforms the baseline at lower resolutions, however, only by a small margin (table 3). This indicates that PACO is not solvable with resolution lower than its full-HD or higher. The performance increase at higher pixel resolutions (that is, 900 and 1400) comes at 1.2x and 1.5x higher latency respectively, compared to the 640 pixel resolution. This shows that, for harder detection tasks, increased resolution does meaningfully improve performance, and methods such as ours can take advantage of the additional detail without a large increase in computational cost.

### 5.3 SCALING BEHAVIOUR OF COMPUTATIONAL COST WITH INCREASING IMAGE RESOLUTION

For this experiment we measured inference latency as a function of input resolution on the Ego4D gaze estimation benchmark, without any additional training. We compared HiResNets to the baseline SqueezeTime, running both models on an NVIDIA M40 GPU. Latency was recorded while

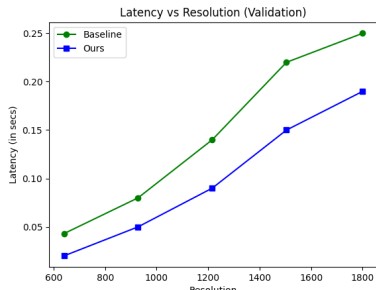

Figure 5: Computational cost plot (latency vs. resolution) on Ego4D (Grauman et al., 2022) object detection task for the baseline (YOLOv5) and our method in the validation setting.

| Method | Max Res | Train | Eval. | AP. | AR. |
|---|---|---|---|---|---|
| | 640 | 0.44 | 0.37 | **0.58** | 0.10 |
| YOLOv5 (Jocher et al., 2020) | 900 | 0.48 | 0.38 | 0.52 | 0.12 |
| | 1400 | 0.55 | 0.43 | 0.55 | 0.18 |
| | 640 | 0.49 | 0.40 | 0.51 | 0.20 |
| Ours | 900 | 0.52 | 0.44 | 0.54 | 0.25 |
| | 1400 | **0.58** | **0.47** | 0.57 | **0.30** |

Table 3: Object Parts Detection on PACO Dataset.

scaling the input from a typical ResNet resolution (224px) up to 1K, using identical preprocessing and batch (size 24) settings.

**Results.** The results, shown in Fig. 3 and 5, show that HiResNets scale much more favorably with resolution. SqueezeTime and YOLOv5 exhibits the expected quadratic growth in latency as image size increases, reflecting the pixel-wise scaling of convolutional layers in a ResNet-style backbone. In contrast, HiResNets grow only nearly linearly with resolution, consistent with their logarithmic–squared property. As a result, HiResNets remain efficient even at 1K input, while SqueezeTime becomes substantially slower.

## 6 ABLATIONS

We experiment with different configurations that result from predicting fresh centers for the transform through the network's depth, keeping center fixed, and varying the input's radius and angle sizes for center prediction. We experiment with increasing the center predictions, by predicting twice (instead of once) for stacks of residual blocks (greater than 4), and observe an increase in evaluation accuracy by 1.5%. This is expected due to the representations drift in later blocks from that of the initial residual block. We also ablate by predicting the center only once for the entire network, which results in accuracy degradation ($> 7\%$). Further, we ablate with using ground-truth image center (no prediction) which leads to an even severe performance hit. This is because our method relies on dynamic changes in fixations / saccades sites (from the updated center predictions), essential for guiding the feature extractor's representations to converge in the regions of interest for stable task prediction. For the polar radius/angle sizes, we experiment with multiple configurations such as keeping the factor of reduction the same (eg.$1/4$) across blocks, resulting in an increase in accuracy of 1%. This is expected due to an increased coverage in the local area of interest (i.e. the high resolution site).

## 7 CONCLUSION

In this work we propose HiResNets, a residual architecture that integrates log-polar warps into the residual stream to enable efficient foveated processing. Our experiments show that HiResNets can mimic gaze allocation, handle inputs at substantially higher resolutions than conventional baselines, and exhibit favourable scaling behaviour with respect to resolution. These properties lead to superior performance on egocentric video object detection, where capturing fine-grained details is critical. Overall, our results highlight foveated representations as a promising direction for building scalable high-resolution vision models without incurring quadratic computational cost.

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
