# OpenReview forum: "HiResNets: Native Full-HD Video Recognition with Foveal Residual Streams"
_ICLR.cc/2026/Conference — ICLR 2026 Conference Desk Rejected Submission_

### Official Review · Reviewer_24A8 · 2025-10-30

**Soundness:** 1
**Presentation:** 2
**Contribution:** 1
**Rating:** 2
**Confidence:** 3

**Summary:**

The manuscript proposes HiResNet to optimize the usage of memory in image and video recognition that integrates log-polar warps into the residual stream in order to enable efficient foveated processing. It uses residual architectures as high-resolution buffers, to which convolutional blocks only read and write via log-polar image warp operations.
A complete high-resolution representation is built up in the residual stream.
A differentiable log-polar warp mechanism enabling adaptive foveated processing is built inside the backbone itself. This allows emphasizing local detail while compressing distant regions.

**Strengths:**

* Optimization of deep learning models in order to process images and video more efficient while also displaying the information appropriately.

**Weaknesses:**

* The manuscript lacks a theoretical framework.
* The HiResNet architecture is superficially described and lacks a diagram.
* There are not enough experimental results.
* There are no visual results on images.

**Questions:**

Can the approach be extended to other networks than those adopted? For example can be used for transformers or video transformers?

---

> ### Author Response · Authors · 2025-12-01
>
> Thank you for your thoughtful review and for recognizing the importance of our work. We address the weaknesses and questions below.
>
> > **W1.** The manuscript lacks a theoretical framework.
>
> We have presented the framework in section 4 covering log-polar warps and computational analysis.
>
> > **W2.** The HiResNet architecture is superficially described and lacks a diagram.
>
> We have strived to present a thorough description in section 4 and a diagram in Figure 1, so would appreciate pointers to understand any concerns.
>
> > **W3.** There are not enough experimental results.
>
> Section 5 covers our experimental results, and we have updated the main paper with more results (highlighted).
>
> > **W4.** There are no visual results on images.
>
> Visual results are presented in figure 1.
>
> > **Q1.** Can the approach be extended to other networks than those adopted? For example can it be used for transformers or video transformers?
>
> This architecture relies on the translational equivariance of residual architecture, hence it would not directly extend to transformers or hybrid CNN-ViT designs.

---

### Official Review · Reviewer_5M8C · 2025-10-30

**Soundness:** 2
**Presentation:** 1
**Contribution:** 3
**Rating:** 4
**Confidence:** 3

**Summary:**

This paper introduces **HiResNets**, a residual network architecture that integrates **log-polar foveated warping** directly into the backbone for efficient **Full-HD video recognition**.
The key idea is to treat the residual stream as a persistent high-resolution buffer, while convolutional blocks operate on a warped log-polar space.
This yields **log-squared (≈ O(log² HW))** computational scaling instead of the usual quadratic growth.
HiResNets predict adaptive focus centres (“saccades”) per block and integrate warped features back into the residual stream.
Experiments on **Ego4D**, **EGTEA**, **EgoObjects**, and **PACO** show improved efficiency and competitive performance for egocentric video recognition.

**Strengths:**

- **Novel idea:** Embeds foveated processing *within* the residual path, rather than as an external module.
- **Theoretical clarity:** Provides a concrete complexity reduction from O(HW) to O(log H log W).
- **Biological motivation:** Well-aligned with foveation and saccade mechanisms in human vision.
- **Efficiency:** Demonstrates reduced latency and better scaling with input resolution.

**Weaknesses:**

1. **Limited comparisons** – Experiments mainly against YOLOv5 and SqueezeTime; no comparison with modern high-resolution backbones (e.g., Swin, MViTv2, FoveaTer).
2. **Modest empirical gains** – Accuracy improvements are small (≈ +1–3 %) relative to baseline.
3. **Implementation cost** – Warp/unwarp overhead not fully analyzed; runtime breakdown missing.
4. **Shallow ablation** – Only evaluates fovea frequency and polar radius; missing tests for fixed centres, inverse warp, or focus visualization.
5. **Reproducibility** – Training details for Full-HD experiments and memory optimization are insufficient.

**Questions:**

- How stable is the training when saccade positions change dynamically?
- Do the predicted focus points align with human gaze or motion saliency?
- Could this architecture extend to transformers or hybrid CNN–ViT designs?
- How efficient are the warp/unwarp operations on GPU compared to standard convolutions?

---

> ### Author Response · Authors · 2025-12-01
>
> Thank you for your thoughtful review and for recognizing the importance of our work. We address weaknesses and questions below in two separate comments.
>
>
> > **W1.** Limited comparisons – Experiments mainly against YOLOv5 and SqueezeTime; no comparison with modern high-resolution backbones (e.g., Swin, MViTv2, FoveaTer).
>
> We have added comparisons with MviTv2 in Table 1.
>
> > **W2.** Implementation cost – Warp/unwarp overhead not fully analyzed; runtime breakdown missing.
>
> We describe computational analysis in section 4.3, and briefly below.
>
> A standard residual block with kernel size $k$ applied to a feature map of size $(W, H)$ has cost $\mathcal{O}(k^2 W H)$. In our proposal, convolutions act only on the warped view $u \in \mathbb{R}^{C \times \Theta \times \rho}$, where the log-polar warp yields $\Theta = \mathcal{O}(\log W)$ and $\rho = \mathcal{O}(\log H)$. The warp itself can be implemented in $\mathcal{O}(\Theta \rho)$, so the dominant cost is the convolution, now logarithmic-square scaling. It replaces the quadratic growth of conventional blocks, and becomes the bottleneck whenever convolution dominates warp overhead and channel dimensions remain moderate.
>
> Meanwhile, the residual stream ensures that no fine detail is lost: every update is reintegrated into a global high-resolution buffer.
> Thus HiResNets achieve logarithmic-square scaling in the computationally dominant convolutional bottlenecks, while preserving complete spatial information across depth and time.
>
>
> > **W3.** Shallow ablation – Only evaluates fovea frequency and polar radius; missing tests for fixed centres, inverse warp, or focus visualization.
>
> We have added the suggested experiments in section 6 (highlighted).
>
> > **W4.**  Reproducibility – Training details for Full-HD experiments and memory optimization are insufficient.
>
> Thank you for pointing, we will add further training details in the final version.
>
>
> > **Q1.** How stable is the training when saccade positions change dynamically?
>
> Throughout the training, the predicted gaze centers go through dynamic changes in fixations and saccades sites. These dynamic changes are essential for downstream task prediction, as it guides the feature extractor to converge in the regions of interest for stable detection and gaze estimation. So, as a result, training is stable when the saccade positions change dynamically.
>
> > **Q2.** Do the predicted focus points align with human gaze or motion saliency?
>
> Yes the predicted focus points align with human gaze and motion saliency which results in stable detection and gaze estimation, as seen in the experimental results (Tables 1-3), and qualitative results in Figure 1.
>
> We will also add some representative qualitative results for this.
>
> > **Q3.** Could this architecture extend to transformers or hybrid CNN–ViT designs?
>
> This architecture relies on the translational equivariance of residual architecture, hence it would not directly extend to transformers or hybrid CNN-ViT designs.
>
> > **Q4.** How efficient are the warp/unwarp operations on GPU compared to standard convolutions?
>
> We address this under W2 with computational analysis. We will also add more comparative metrics to show this efficiency in the final version.

---

### Official Review · Reviewer_4ukr · 2025-11-01

**Soundness:** 2
**Presentation:** 1
**Contribution:** 2
**Rating:** 2
**Confidence:** 3

**Summary:**

The paper proposes logarithmic-square growth in resolution scaling, instead of quadratic scaling. This enables processing native videos efficiently, which was prohibitive for standard CNNs. A novel architecture with biologically foveating inside the backbone is proposed. The algorithm seems to learn meaningful foveating strategies and shows performance on challenging tasks, such as gaze estimation, object detection, and fine-grained recognition. The algorithm may be good at handling very small objects and is adaptive to multiple scales.

**Strengths:**

- Proposes a new scaling to deal with full-resolution videos, which is innovative, making the algorithm useful and effective
- Shows a sophisticated design for foveating inside a backbone that is inspired by biological vision, and is validated on benchmark datasets
- Shows results on Ego4D and EgoObjects and PACO datasets

**Weaknesses:**

- It seems a bit unclear whether the algorithm has been tested on datasets with a large enough scale. Perhaps experiments on well-known large-scale detection benchmarks COCO, ImageNet will present a boost to its impact
- More comprehensive ablation and analysis will be necessary for the proposed claims to be convincing, e.g. in which situations is the foveating effective, are there validation experiments for resolution scaling, and its effectiveness in hardware
- The paper could be better presented and written, e.g., with improved writing, more compact tables/figures

**Questions:**

- I wonder if the method could be tested on imagenet or COCO, even a smaller subset to compare with sota results
- Could more ablation studies be added to the paper and verify claims?
- better writing and table/experiment result presentation will make the paper stronger

---

> ### Author Response · Authors · 2025-12-01
>
> Thank you for your thoughtful review and for recognizing the importance of our work. We jointly address weaknesses and questions below.
>
> > **W1, Q1.** It seems a bit unclear whether the algorithm has been tested on datasets with a large enough scale. Perhaps experiments on well-known large-scale detection benchmarks COCO, ImageNet will present a boost to its impact
>
> We have tested our proposed method on large-scale datasets like Ego4D (6 million images) which is larger than COCO (300K images) and Imagenet (1.2 million images).
>
> > **W2, Q2.** More comprehensive ablation and analysis will be necessary for the proposed claims to be convincing, e.g. in which situations is the foveating effective, are there validation experiments for resolution scaling, and its effectiveness in hardware
>
> We address these concerns by adding the suggested experiments (resolution scaling and effectiveness foveation) in section 5.2 and 5.3 (Figure 5). We have also added new ablations in section 6 (highlighted).
>
> > **W3, Q3.** The paper could be better presented and written, e.g., with improved writing, more compact tables/figures.
>
> We have improved clarity, and would appreciate any points to improve writing and clarify parts that are unclear. We will also condense the tables and figures in the final version.

---

### Official Review · Reviewer_JTLt · 2025-11-01

**Soundness:** 2
**Presentation:** 2
**Contribution:** 2
**Rating:** 4
**Confidence:** 3

**Summary:**

The paper proposes HiResNets, using a Foveal Residual Stream and a new convolution to enable efficient Full-HD video recognition. The method is inspired by human vision, focusing high-resolution computation on important regions while keeping low-resolution global context. Experiments show good accuracy and efficiency.

**Strengths:**

* The paper tackles the high cost of scaling vision models to Full-HD, proposing a method that reduces complexity.
* The “Foveal Residual Stream” is an nice idea to put more attention on  high-res focus where it matters while keeping global low-res context.
* The results show good performance

**Weaknesses:**

*  It’s hard to tell whether the efficiency comes from, which makes the technical novelty less convincing.
* The proposed method lacks strong theoretical support and feels like a task-specific trick rather than a general tool.

**Questions:**

* Could the authors do more fine-grained ablation study and how the effectiveness of the propose approach?

---

> ### Author Response · Authors · 2025-12-01
>
> Thank you for your thoughtful review and for recognizing the importance of our work. We address weaknesses and questions below.
>
>
> > **Weakness 1.**  It’s hard to tell whether the efficiency comes from, which makes the technical novelty less convincing.
>
> We have presented the computational analysis in section 4.6.
>
> > **Weakness 2.**  The proposed method lacks strong theoretical support and feels like a task-specific trick rather than a general tool.
>
> We have presented our theoretical framework in section 4.1 through section 4.5 covering log-polar warps and computational analysis.
>
>
> > **Question 1.**  Could the authors do more fine-grained ablation study and show the effectiveness of the proposed approach?
>
> We have added fine-grained ablations in section 6 (highlighted), and new experiments for effectiveness study in section 5.2 (highlighted), and an additional figure 5 (in section 5.3).

---

### Note · Program_Chairs · 2026-01-17
**Submission Desk Rejected by Program Chairs**

The following references in this submission do not refer to real documents and/or have major errors in bibliographic information:

 Xiang Li et al. Mram: Memory-augmented recurrent attention models. arXiv preprint arXiv:2503.12345, 2025
Yasuyuki Yamada et al. Foveated rendering and its applications to virtual reality. Displays, 52:1-9, 2018.